# Atomic Layer Deposition of aTiO$_2$ Layer on Nitinol and Its Corrosion Resistance in a Simulated Body Fluid

**Rebeka Rudolf** [1,*] , **Aleš Stambolić** [2] and **Aleksandra Kocijan** [2]

1   Faculty of Mechanical Engineering, University of Maribor, Smetanova ulica 17, 2000 Maribor, Slovenia
2   Institute of Metals and Technology, Lepi pot 11, 1000 Ljubljana, Slovenia; ales.stambolic@gmail.com (A.S.); aleksandra.kocijan@imt.si (A.K.)
*   Correspondence: rebeka.rudolf@um.si

**Abstract:** Nitinol is a group of nearly equiatomic alloys composed of nickel and titanium, which was developed in the 1970s. Its properties, such as superelasticity and Shape Memory Effect, have enabled its use, especially for biomedical purposes. Due to the fact that Nitinol exhibits good corrosion resistance in a chloride environment, an unusual combination of strength and ductility, a high tendency for self-passivation, high fatigue strength, low Young's modulus and excellent biocompatibility, its use is still increasing. In this research, Atomic Layer Deposition (ALD) experiments were performed on a continuous vertical cast (CVC) NiTi rod (made in-house) and on commercial Nitinol as the control material, which was already in the rolled state. The ALD deposition of the TiO$_2$ layer was accomplished in a Beneq TFS 200 system at 250 °C. The pulsing times for TiCl$_4$ and H$_2$O were 250 ms and 180 ms, followed by appropriate purge cycles with nitrogen (3 s after the TiCl$_4$ and 2 s after the H$_2$O pulses). After 1100 repeated cycles of ALD depositing, the average thickness of the TiO$_2$ layer for the CVC NiTi rod was 52.2 nm and for the commercial Nitinol, it was 51.7 nm, which was confirmed by X-ray Photoelectron Spectroscopy (XPS) and Scanning Electron Microscope (SEM) using Energy-dispersive X-ray (EDX) spectroscopy. The behaviour of the CVC NiTi and commercial Nitinol with and without the TiO$_2$ layer was investigated in a simulated body fluid at body temperature (37 °C) to explain their corrosion resistance. Potentiodynamic polarisation measurements showed that the lowest corrosion current density (0.16 μA/cm$^2$) and the wider passive region were achieved by the commercial NiTi with TiO$_2$. Electrochemical Impedance Spectroscopy measurements revealed that the CVC NiTi rod and the commercial Nitinol have, for the first 48 h of immersion, only resistance through the oxide layer, as a consequence of the thin and compact layer. On the other hand, the TiO$_2$/CVC NiTi rod and TiO$_2$/commercial Nitinol had resistances through the oxide and porous layers the entire immersion time since the TiO$_2$ layer was formatted on the surfaces.

**Keywords:** nitinol; continuous vertical cast (CVC), NiTi rod; atomic layer deposition; corrosion properties; potentiodynamic test; electrochemical impedance spectroscopy

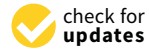



## 1. Introduction

Nitinol is a group of alloys that are in the equiatomic composition range of nickel and titanium. It shows unique properties, such as superelasticity and a Shape-Memory Effect. It also exhibits good corrosion resistance in a chloride environment, an unusual combination of strength and ductility, a high tendency for self-passivation, high fatigue strength, low Young's modulus and excellent biocompatibility. Its properties have enabled its use, especially for biomedical purposes (orthodontic treatments, cardiovascular surgery for stents and guide wires, orthopaedic surgery for various staples and rods, maxillofacial and reconstructive surgery). In addition, Nitinol has been used in the Aerospace, Automotive, Marine and Chemical industries and Civil and Structural Engineering [1–7].

Nitinol has a tendency for self-passivation in a physiological saline solution. The passive films formed on the surface consist mainly of amorphous titanium dioxide [6]. However, the naturally formed oxides on the metal surfaces are thin and do not prevent the corrosion process or Ni leaching. The main problem of Nitinol is its high Ni content. Ni releasing can induce toxic, allergic and hypersensitive reactions or tissue necrosis after long-term implantation [8–10]. Leaching of Ni can arise when a strongly acidic fluid attacks the surface of the alloy. This corrosion is accompanied by nickel release from an implant into the surrounding body fluid and tissue, which can enhance an allergic reaction in a sensitive organism. Another path leading to the accumulation of Ni in the surface layers can be the type of surface treatment itself. Low-temperature (60–160 °C) pre-treatment protocols or high-temperature annealing in the air used for deposition of a thick $TiO_2$ layer onto a Nitinol surface results in Ni accumulation in the surface depth [11–13]. This hidden Ni can easily be released through the defective surfaces, exceeding the Ni release from non-treated material by two to three orders of magnitude. A high concentration of Ni close to the metal–oxide interface will yield larger particles. The larger particles will induce severe local strain in the lower oxide layer, leading to local rupture and cracking. Such cracks can then extend towards the surface and act as channels, explaining the much greater Ni release. To prevent corrosion and, consequently, Ni release, a coating of appropriate thickness must be formed on the NiTi surface. Titanium oxide coatings are useful enough to suppress nickel ions' out-leaching [14–18].

Titanium dioxide is formed when titanium is subjected to oxidising conditions. $TiO_2$ is an electron excess conductor with oxygen deficiency. Due to the electroneutrality condition oxygen vacancies must be compensated by a corresponding number of negative charges. This can be achieved by $O^{2-}$ ions that are incorporated into the lattice, resulting in electroneutrality of the complete crystal. The disorder of the oxide determines which species are mobile in the course of oxidation. The driving force for diffusion is the concentration gradient of the vacancies in the oxide, such as, in $TiO_2$ oxygen ions diffuse inward since the oxygen vacancy concentration is highest at the oxide/metal interface. The disorder in oxides not only determines the location for scale growth, but also influences the ability of the oxide scale to close the cracks by oxide regrowth during high-temperature exposure [19]. Oxidation of Nitinol occurs as follows:

$$NiTi + O_2 \rightarrow Ni_3Ti + TiO_2 \rightarrow Ni_4Ti + TiO_2 \rightarrow Ni + TiO_2 \qquad (1)$$

Due to the four-times lower Gibbs free energy of formation of titanium oxide than that of nickel oxide, titanium oxide growth is preferred on the Nitinol surface. Observations are consistent with a model of oxygen absorption on the NiTi surface that reacts with outward diffusing Ti to form $TiO_2$. During the early stages of oxidation, the growth of the $TiO_2$ layer is the only contributor to thickness, and, therefore, oxide formation is relatively rapid. However, the preferential oxidation of Ti creates a Ti-depleted (Ni-rich) zone at the $NiTi/TiO_2$ interface. The formation of the Ni-rich layer increases the effective diffusion distance with an associated decrease in overall oxidation kinetics. Therefore, continued oxide growth involves the simultaneous nucleation and growth of titanium oxides and Ni-rich phases. Ultimately, these processes lead to the formation of a protective oxide scale, which prevents further oxidation of the base material [20,21]. Naturally grown titanium oxide on a Nitinol surface is approximately 5 nm thick, and thicker titanium oxide should be formed to achieve better properties [22].

The most common methods employed for layer formation are anodisation, plasma spraying, Atomic Layer Deposition (ALD), etc. ALD is an interesting technique for producing $TiO_2$ thin films due to its simplicity, reproducibility, high conformity of thin films and excellent control of the layer thickness at the angstrom level. The thin film is formed as a result of repeated deposition cycles. At least two subsequent self-limited surface reactions are used to form a new layer [23–26]. In the literature, there are results for a commercial flat-annealed NiTi foil and NiTi wires ($\phi$ = 0.2 mm) [27], and for $Al_2O_3$ and Pt, ALD coatings on NiTi thin films [28]. None of the studies, however, dealt with the study of

ALD deposition on the surface of a continuous vertical cast (CVC) NiTi rod ($\phi$ = 11 mm), which was vacuum remelted before casting. Namely, the production of Nitinol is still very complex, and, therefore, new faster processes resulting in the shape of a rod with the smallest diameter possible are being sought.

The ALD technique for producing $TiO_2$ on a CVC NiTi rod made in-house [29] was tested in this study. For comparison, a commercial Nitinol was used to evaluate how the ALD coating works. CVC NiTi rod-testing production is explained in our previous works [30,31], while the commercial Nitinol was already in the rolled state. With the ALD technique, the thin layer of titanium oxide was deposited on both types of specimens. The formation of $TiO_2$ was investigated by X-ray Photoelectron Spectroscopy (XPS) and Scanning Electron Microscopy (SEM) using Energy-dispersive X-ray (EDX) spectroscopy. $TiO_2$-protected specimens were tested on corrosion behaviour in simulated body fluid to clarify their corrosion resistance and confirm the positive effect of the formatted $TiO_2$ layer in order to reduce the corrosion rate.

## 2. Materials and Methods

### 2.1. Atomic Layer Deposition of a $TiO_2$ Layer

The process of depositing the layer onto the surface with ALD is cyclic. Each cycle is set up from 4 steps (see Figure 1). In the first step, the first precursor ($TiCl_4$) is pulsed into the reactor. $TiCl_4$ is adsorbed to the specimen's surface and reacts there with the reactive sites (initially, these are organic impurities). In the second step, nitrogen is pulsed into the reactor to purge the excess of the reactant and by-products. In the third step, the second precursor ($H_2O$) is pulsed into the reactor. $H_2O$ is adsorbed to the specimen's surface and reacts there with the reactive sites. In the last step, nitrogen is again pulsed into the reactor to purge the excess of the reactant and by-products. 1100 cycles were made in this study. The simplified equation for this reaction is:

$$TiCl_4 + 2\,H_2O \rightarrow TiO_2 + 4\,HCl \tag{2}$$

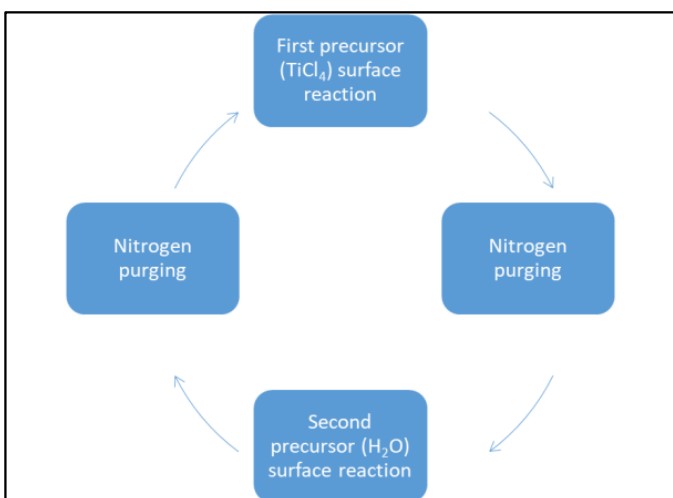

**Figure 1.** Steps of Atomic Layer Deposition (ALD) deposition.

The deposition of a $TiO_2$ layer on the surface of a continuous vertical cast (CVC) NiTi rod and commercial Nitinol was accomplished in a Beneq TFS 200 system at 250 °C. The pulsing times for $TiCl_4$ and $H_2O$ were 250 ms and 180 ms, followed by appropriate purge cycles with nitrogen (3 s after the $TiCl_4$ and 2 s after the $H_2O$ pulses). The chemical compositions of specimens were as follows: Commercial Nitinol (45 wt.% Ti, 55 wt.% Ni) and the CVC NiTi rod made in-house (composition determined by XRF analysis in-house: Ti 38.9 wt.%, 59.8 wt.% Ni). The detailed microstructure investigation revealed that the CVC

NiTi rod containing over 50 at. % Ni, consisted of $Ti_2Ni$ and cubic NiTi, with corresponding EDX spectra in the field of investigation—see Figure 2. The chemical composition of the CVC NiTi rod varied through the cross and longitudinal sections because the drawing process was not optimal; the manufacturing problem is described in more detail in our previous study [29].

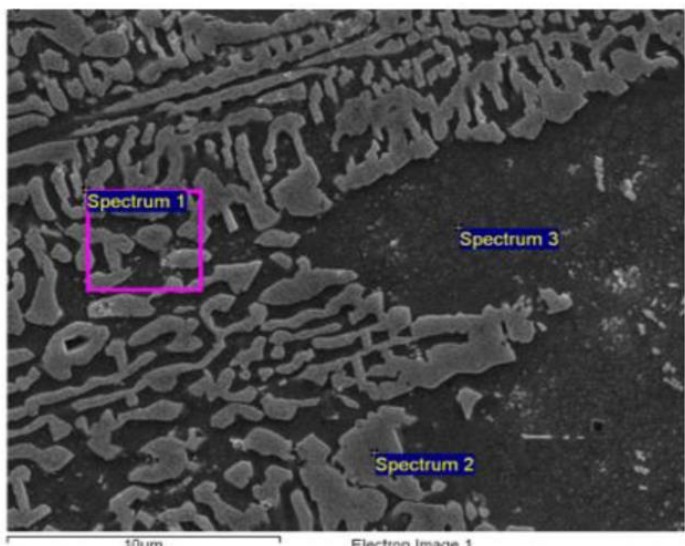

| EDX· Spectrum | Ti·(in·at.%) | Ni·(in·at.%) |
|---|---|---|
| 1 | 38.38 | 61.62 |
| 2 | 31.68 | 68.32 |
| 3 | 40.14 | 59.86 |
| | | |
| mean | 36.73 | 63.26 |
| max | 40.14 | 68.32 |
| min | 31.68 | 59.86 |

**Figure 2.** Microstructure of the continuous vertical cast (CVC) NiTi rod with corresponding Energy-dispersive X-ray (EDX) spectrum.

The X-ray Photoelectron Spectroscopy (XPS) analyses were performed in order to identify the oxidation states of the elements on the surface of the Ti-oxide films and calculate their surface composition. XPS analyses were carried out on the PHI-TFA XPS spectrometer (Physical Electronics Inc., MN, USA), equipped with a monochromatic Al source. The analysed area was 0.4 mm in diameter, and the analysed depth was about 3–5 nm. High energy resolution XPS spectra were taken with a pass energy of 29 eV, energy resolution of 0.6 eV and energy step of 0.1 eV. Quantification of the surface composition was performed from the XPS peak intensities, taking into account the relative sensitivity factors provided by the instrument manufacturer [32]. Two places on every specimen were analysed, and the average composition was calculated. Chemical bonding of the elements was deduced from the high-energy resolution XPS spectra using reference XPS1. The Multipak software package (version 9.9, ULVAC-PHI Inc., Japan) was used for XPS spectra processing. The error in the binding energy of the measured spectra is ±0.3 eV.

Additionally, a Jeol JSM-7800F Field Emission SEM, equipped with EDX spectroscopy X-Max[N] 80, Oxford Instruments (Oxford Instruments, Abingdon, UK), was used for identification of the EDX line chemical composition of the formatted $TiO_2$ analysis on the surface

of the CVC NiTi rod and commercial Nitinol. For this purpose, the CVC NiTi rod was cut into a cylinder shape with $\phi$ = 11 mm and h = 1 cm, using an Accutom 50 (IMT, Ljubljana, Slovenia) electronic saw for precision cutting. The commercial Nitinol, which was in the rolled state in the form of sheets with width of 1.5 mm, was cut to circles with $\phi$ = 10 mm and a width of 1.5 mm using a water jet cutter (Faculty of Mechanical Engineering, Ljubljana, Slovenia) (Omax, Kent, WA, USA). For easier polishing, the specimens were then hot-pressed into Bakelite. Mechanical polishing was performed on Struers Abramin apparatus (IMT, Ljubljana, Slovenia) (Struers, Copenhagen, Denmark). The grinding was performed with 320-grit SiC abrasive paper, mechanical polishing with MD-Largo (Struers, Cleveland, OH 44145, USA) discs with 9 μm diamond suspension and with peroxide grains in a chemically aggressive suspension—OP-S (colloidal silica). At the end, specimens were cleaned with detergent, washed well with water and put in an ultrasound bath in alcohol. Surface images were recorded at 20,000× magnification, where the SEM working parameters were 0.7 kV voltage and 2 mm working distance. Cross-section images were recorded at 200,000× magnification and SEM working parameters of 10 kV voltage and 4 mm working distance.

### 2.2. Corrosion Tests

The corrosion tests were performed on the CVC NiTi rod with and without a formatted $TiO_2$ layer and on commercial Nitinol with and without a formatted $TiO_2$ layer, respectively. All the measurements were held at body temperature (37 °C). Potentiodynamic polarisation measurements and Electrochemical Impedance Spectrometry (EIS) (Biologic, Seyssinet-Pariset, France) were used to study the electrochemical behaviour of the specimens. All the measurements were recorded by a BioLogic Modular Research Grade Potentiostat/Galvanostat/FRA Model SP-300 (Seyssinet-Pariset, France) with an EC-Lab Software (V 11.27, Biologic, Seyssinet-Pariset, France) and three-electrode cell. In this cell, the specimen, with an exposed area of 1 cm$^2$, was a working electrode, and a saturated calomel electrode (SCE, 0.242 V vs. SHE) was used as a reference electrode, and the Counter Electrode (CE) was a platinum net. The experiment was held in simulated physiological Hank's solution, containing 8 g/L NaCl, 0.40 g/L KCl, 0.35 g/L NaHCO$_3$, 0.25 g/L NaH$_2$PO$_4$ × 2H$_2$O, 0.06 g/L Na$_2$HPO$_4$ × 2H$_2$O, 0.19 g/L CaCl$_2$ × 2H$_2$O, 0.41 g/L MgCl$_2$ × 6H$_2$O, 0.06 g/L MgSO$_4$ × 7H$_2$O and 1 g/L glucose, at pH = 7.8. All the chemicals were from Merck, Darmstadt, Germany. The potentiodynamic curves were recorded after 1 h specimen stabilisation at the Open-Circuit Potential (OCP), starting the measurement at 250 mV vs. SCE more negative than the OCP. The potential was then increased, using a scan rate of 1 mV s$^{-1}$ until the transpassive region was reached. Long-term open-circuit potentiostatic electrochemical impedance spectra were obtained for the investigated specimens. The impedance was measured at the OCP, with sinus amplitude of 5 mV peak-to-peak and a frequency range of 65 kHz to 1 mHz, in the sequence of 0 h, 1 h, 2 h, 6 h, 12 h, 24 h, 48 h, 72 h, 96 h, 120 h, 144 h, 168 h and 192 h. The impedance data are presented in terms of Nyquist plots. Zview v3.4d Scribner Associates software (V 3.4d, Southern Pines, NC, USA) was used for the fitting process. All the experiments were repeated 3 times.

## 3. Results and Discussion

### 3.1. ALD $TiO_2$ Thin Layer Deposition

SEM investigation revealed that after 1100 repeated ALD cycles, the average thickness of the $TiO_2$ layer for the CVC NiTi rod was 52.2 nm, and for the commercial Nitinol 51.7 nm, as visible in Figure 3.

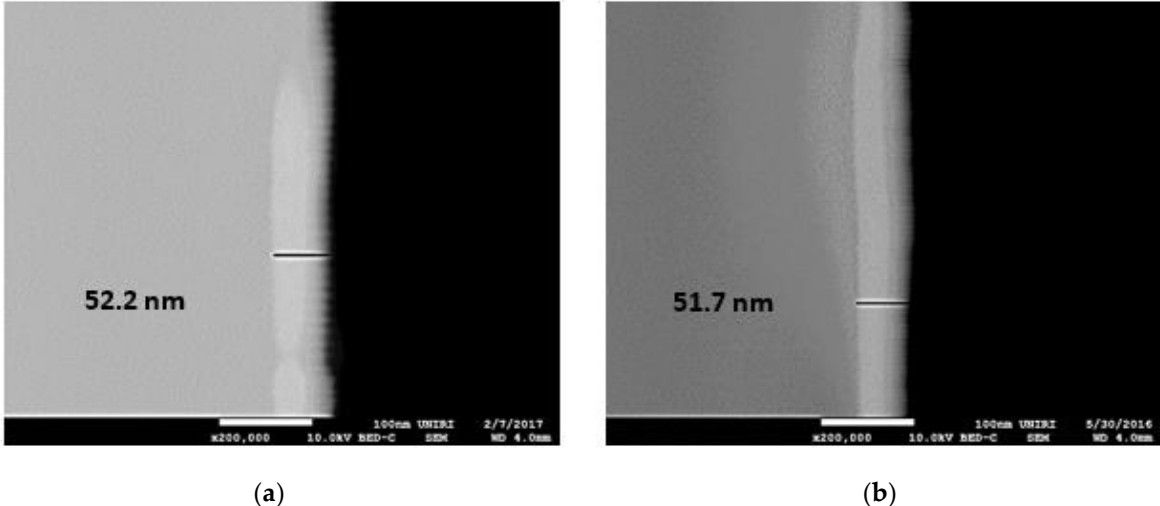

(**a**) (**b**)

**Figure 3.** Scanning electron microscopy (SEM) image of the TiO$_2$ layer at 200,000× magnification of the cross-section for the (**a**) CVC NiTi rod and (**b**) Commercial Nitinol.

As seen in Figure 4, the deposited TiO$_2$ layer is quite similar on both surfaces (CVC NiTi rod and commercial Nitinol). The particles are quite different in size (from 50 nm to 1 μm), while their shape is fairly tetrahedral with quite pointed edges, and the grain boundaries are clearly visible.

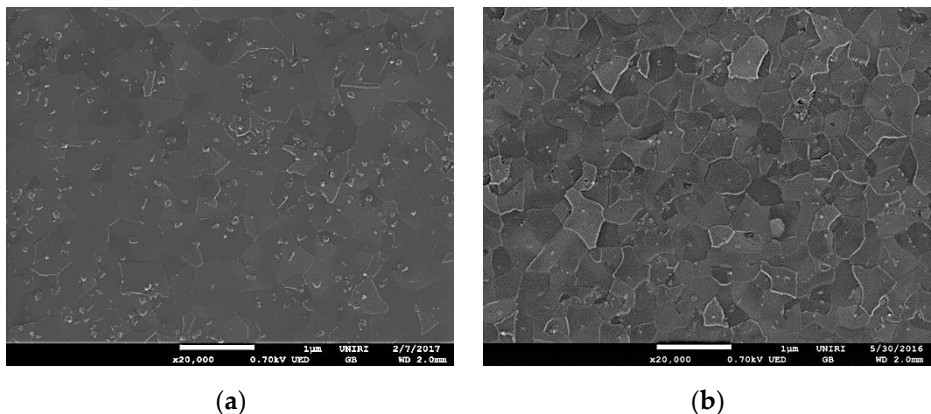

(**a**) (**b**)

**Figure 4.** SEM image at 20,000× magnification of a TiO$_2$ layer on the surface of: (**a**) CVC NiTi rod and (**b**) Commercial Nitinol.

Figure 5a shows an XPS survey spectrum from the TiO$_2$/com.NiTi, and Figure 5b shows a survey spectrum from the TiO$_2$/CVC NiTi rod. Both spectra are similar. They contain the following peaks: Ti 2p$_{3/2}$ at 458.6 eV, C 1s at 284.8 eV, O 1s at 530.0 eV, Ti 3p at 38.2 eV, Ti 3s at 62.5 eV, N 1s at 401.0 eV, O KLL Auger peak at 974 eV and Ti LMM Auger peak at 1106 eV. No traces of a Cl 2p peak were found, which would be expected at 198 eV. From Ti 2p, O 1s, C 1s and N 1s spectra, a surface composition for TiO$_2$/com. NiTi was calculated to be: 45.6 at.% of O, 17.9 at.% of Ti, 35.7 at.% of C and 0.9 at.% of N. Surface composition for the TiO$_2$/CVC NiTi rod was: 42.0 at.% of O, 16.5 at.% of Ti, 40.4 at.% of C and 1.1 at.% of N. The O/Ti ratio for both specimens was the same, i.e., 2.5. The presence of carbon atoms and part of the oxygen atoms are probably related to surface contamination and/or specimen preparation, taking into account that part of the oxygen is related to surface contamination. The Ti/O ratio of 2.5 indicated the TiO$_2$ composition of the deposited films on both specimens. XPS data on the surface composition show that the TiO$_2$/CVC NiTi rod had a higher concentration of carbon, which may be related to the higher degree of surface contamination.

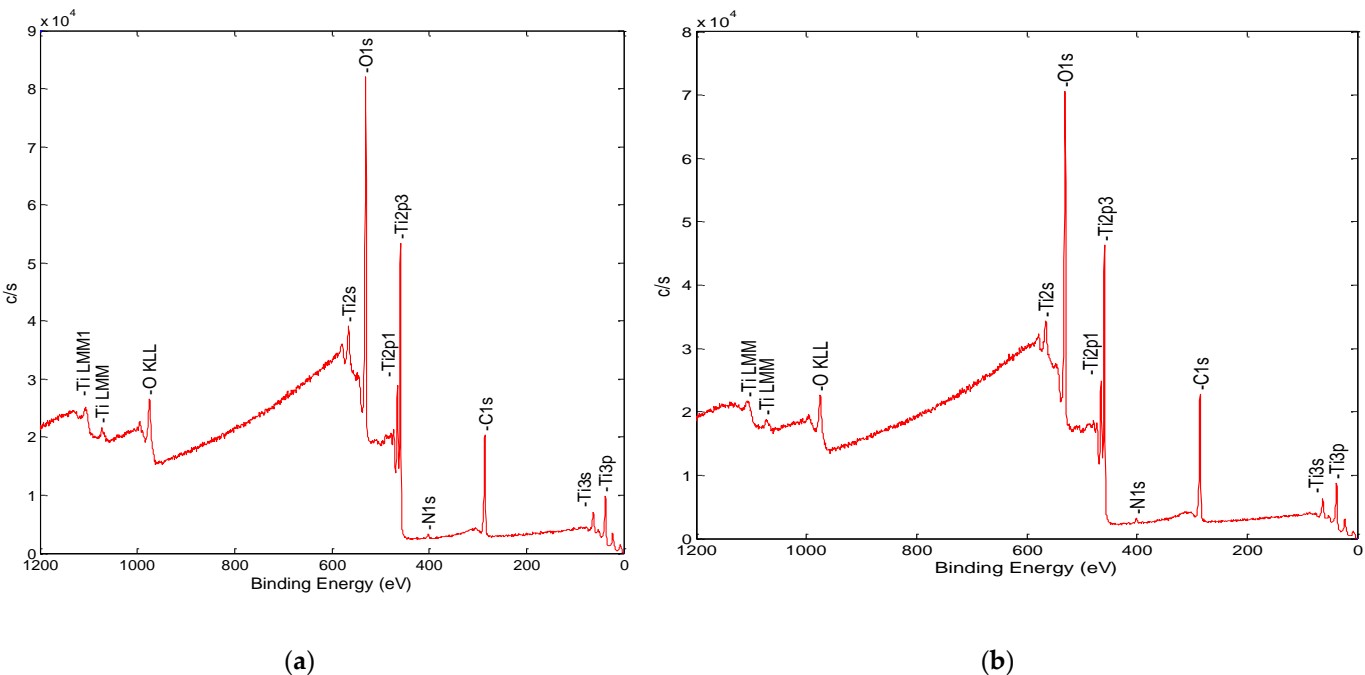

**Figure 5.** XPS survey spectra from (**a**) the TiO$_2$/com.NiTi and (**b**) the TiO$_2$ CVC NiTi rod.

In order to get an insight into the surface chemistry, high-energy resolution XPS spectra of Ti 2p, O 1s and C 1s were acquired on both specimens. The high-energy resolution XPS spectra Ti 2p, O 1s and C 1s from TiO$_2$/com.NiTi were deconvoluted into different chemical components and are shown in Figure 6. The high-energy resolution XPS spectra Ti 2p, O 1s and C 1s from the TiO$_2$/CVC NiTi rod were deconvoluted into different chemical components and are shown in Figure 7. It is visible that the high-energy resolution XPS spectra from both specimens are very similar. On the surface of both specimens, the Ti 2p$_{3/2}$ peak is at 458.6 eV, and the Ti 2p$_{1/2}$ peak is at 464.5 eV. This binding energy is related to the Ti(4+) oxidation state, which shows the presence of the TiO$_2$ compound (Figures 6a and 7a). No presence of Ti was identified in the lower oxidation states, at least inside the sensitivity of the XPS method. The Ti 2p$_{3/2}$ peak was very narrow, indicating an ordered TiO$_2$ film. The Full Width at Half Maximum (FWHM) of the Ti 2p$_{3/2}$ peak for both specimens was 1.07 eV. The oxygen spectra O 1s were composed of three peaks, the O1 peak at 530.0 eV, the O2 peak at 531.3 eV and a small peak O3 at 532.6 eV. The O1 peak is related to the O$^{2-}$ anions in the TiO$_2$ lattice. The O2 peak at 531.3 eV may be related to the presence of surface OH-groups or O-vacancies in the oxide. The O3 peak may be related to H$_2$O and/or C-O species, mainly due to surface contamination. The carbon C 1s spectra from both specimens contained four peaks, which were related with different chemical bonds of the carbon atoms: the peak at 284.8 eV (C-C/C-H bonds), the peak at 286.2 eV (C-O/C-OH), the peak at 287.1 eV (O-C-O/C = O) and the peak at 289.5 eV (O = C-O/CO3). XPS results show that the surface of the Ti layer was covered by TiO$_2$ with some surface contamination.

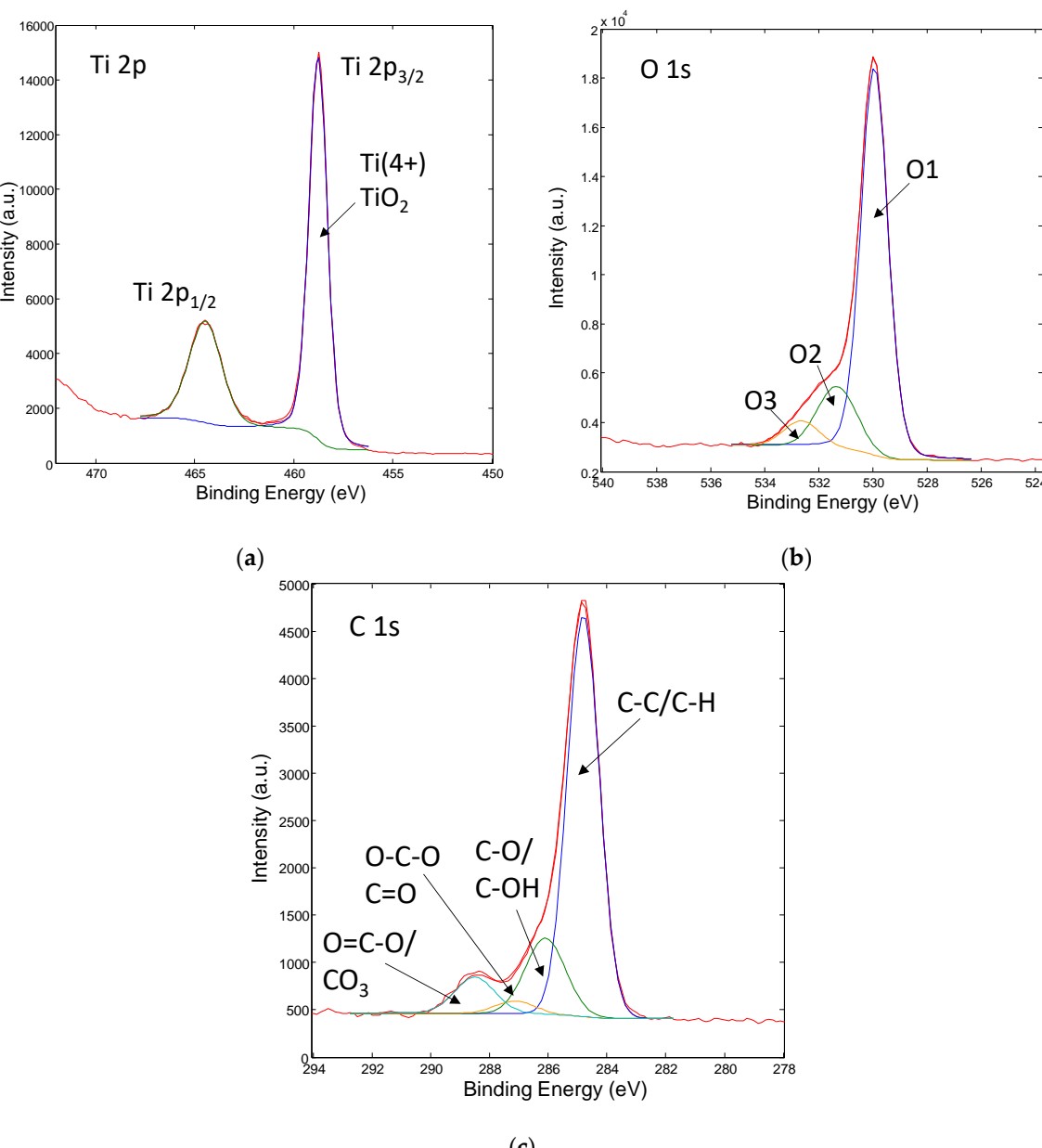

**Figure 6.** High-energy resolution XPS spectra from the $TiO_2$/com.NiTi surface for: (**a**) Ti 2p, (**b**) O 1s and (**c**) C 1s.

In accordance with the XPS analyses, the results of the performed line EDX analysis on the formatted $TiO_2$ layer confirmed the increased contents of Ti and O (Figure 8) in both specimens, exactly where the $TiO_2$ layer formed. The $TiO_2$ layer is seen as a bright area. The increased concentration profiles for Ti and O are appropriate, as well as the width of the range corresponding to the resulting thickness of the $TiO_2$ layer, as seen with SEM (Figure 3). Due to the extremely thin $TiO_2$ layer, only line EDX analysis could be used in this case, as the usual point/planar EDX analysis would capture a significantly larger volume, and in this way, the signal from the base alloy would be obtained, and the results would be irrelevant.

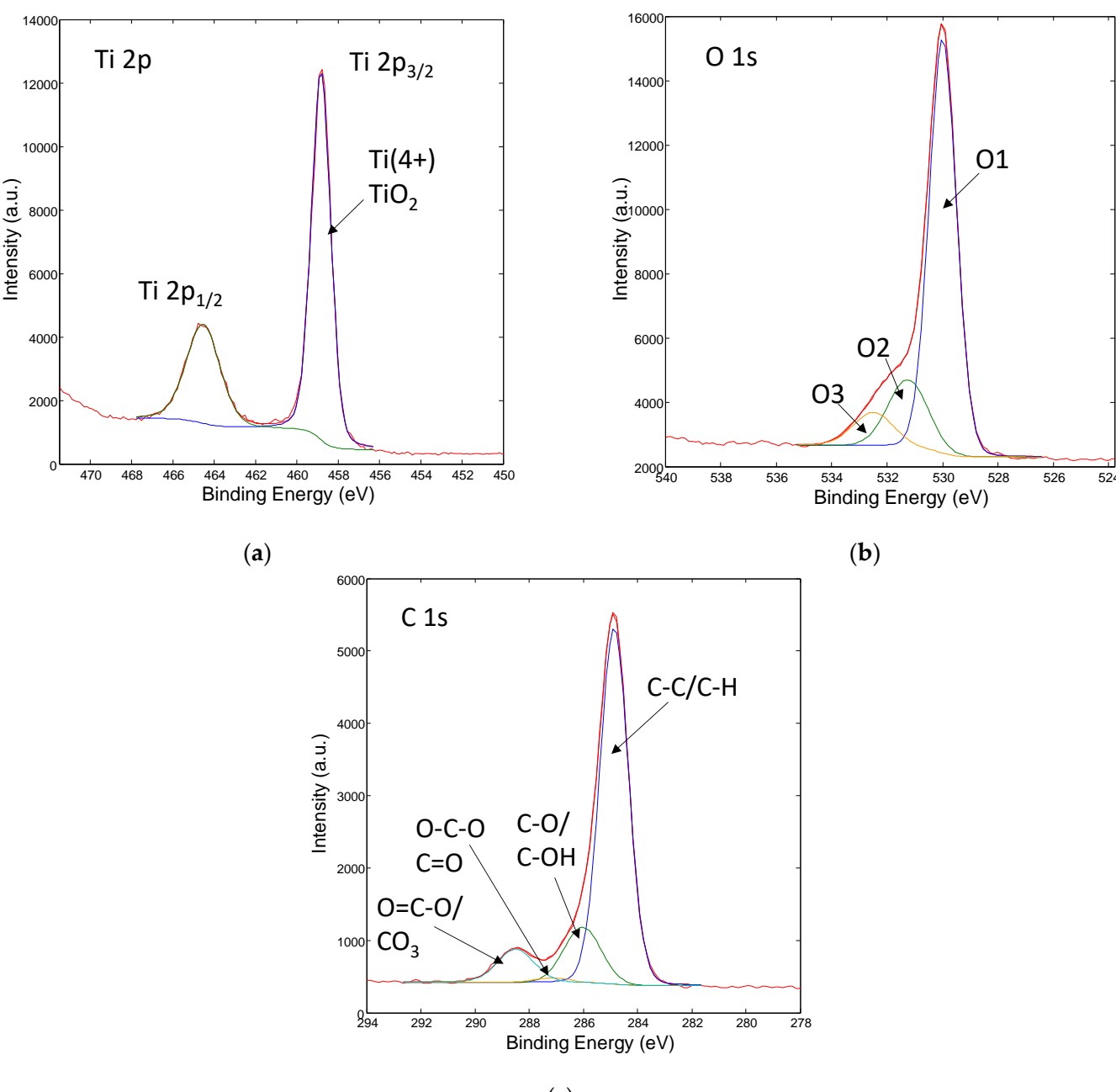

**Figure 7.** High-energy resolution XPS spectra from the the TiO$_2$ CVC NiTi rod surface for: (**a**) Ti 2p, (**b**) O 1s and (**c**) C 1s.

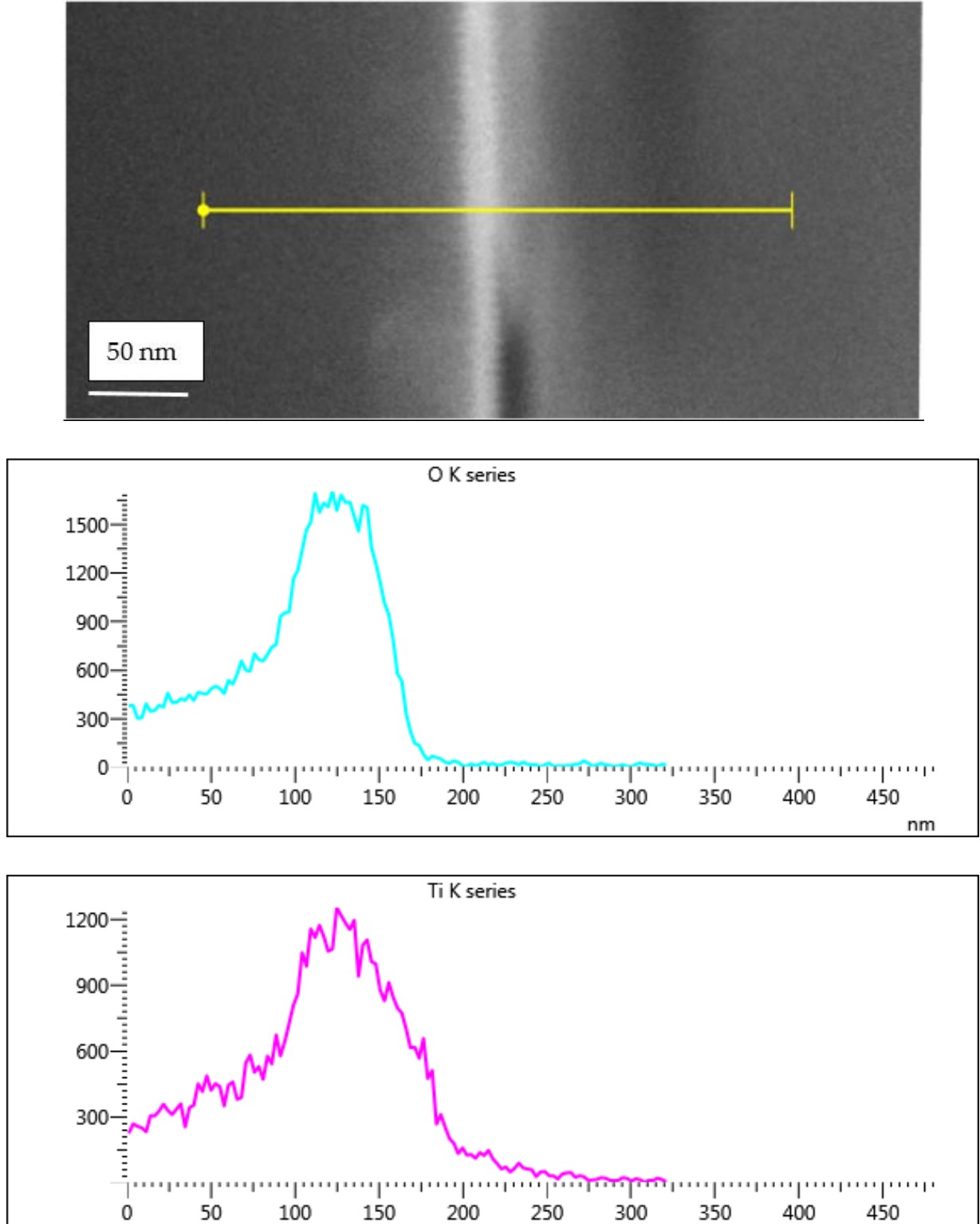

**Figure 8.** EDX linescan of Ti and O in the area of the formatted TiO$_2$ (CVC NiTi rod—the environment is on the left side).

### 3.2. Corrosion Tests

### 3.2.1. Potentiodynamic Test

Figure 9 shows the potentiodynamic curves for different specimens: CVC NiTi rod, commercial Nitinol (com. NiTi), CVC NiTi rod with a TiO$_2$ layer (TiO$_2$/CVC NiTi rod) and commercial Nitinol with a TiO$_2$ layer (TiO$_2$/com NiTi). All the measurements were carried out in Hank's solution at body temperature (37 °C). The electrochemical parameters of the potentiodynamic test are shown in Table 1. Corrosion potentials ($E_{corr}$) and corrosion current densities ($i_{corr}$) were obtained from the Tafel region. Following that region, the specimens exhibited a passive region, which was limited by the breakdown potential ($E_{bd}$), corresponding to the transpassive oxidation of metal species. The corrosion current density was the lowest for TiO$_2$/com NiTi (0.16 µA/cm$^2$), which means that the passive layer on these specimens was the most stable and resistant to external influences. This is followed by specimens of com. NiTi (0.30 µA/cm$^2$), TiO$_2$/CVC NiTi rod (0.34 µA/cm$^2$) and CVC NiTi rod (0.44 µA/cm$^2$). The formation of a passive layer is characterised by the width of the passive region. The wider the passive region, the more corrosion-resistant the material is. The CVC NiTi rod had the smallest passive range has a (from −100 mV to 330 mV, which is 430 mV), followed by TiO$_2$/CVC NiTi rod (650 mV), com. NiTi (730 mV) and TiO$_2$/com. NiTi. The CVC NiTi rod had the lowest breakdown potential (329 mV), followed by com. NiTi (634 mV) and TiO$_2$/CVC NiTi rod (643 mV), while TiO$_2$/com. NiTi did not reach the breakdown potential in the measurement range. As the name already suggests, this passive layer breaks down at this potential, which enables the formation of pits on the surface. This is also accompanied by a rapid increase in anode current due to the passivity breakdown. The corrosion rate is, thus, the smallest for TiO$_2$/com. NiTi; the TiO$_2$/CVC NiTi rod and com. NiTi are far away but very close together, while the CVC NiTi rod shows the highest corrosion rate or the lowest corrosion resistance by the potentiodynamic test.

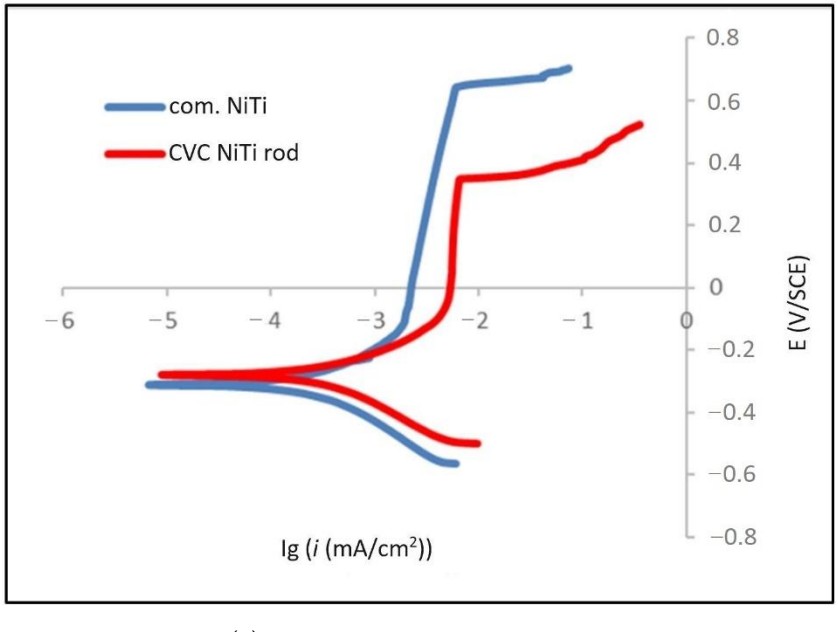

(**a**)

**Figure 9.** *Cont.*

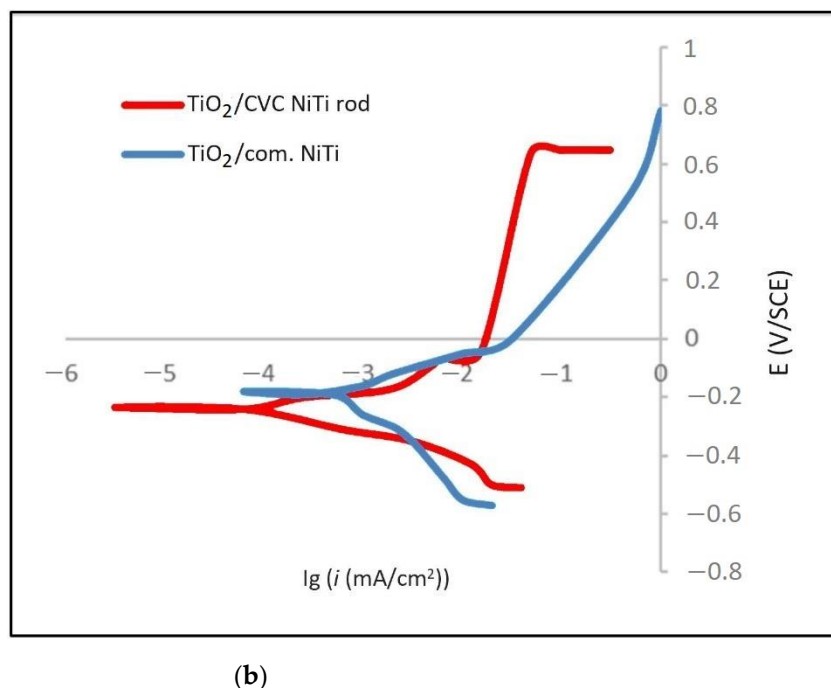

(**b**)

**Figure 9.** Potentiodynamic curves for: (**a**) CVC NiTi rod and com. NiTi and (**b**) TiO$_2$/CVC NiTi rod and TiO$_2$/com. NiTi.

**Table 1.** Electrochemical parameters determined from the potentiodynamic curves.

| Sample | $E_{corr}$ (mV) | $i_{corr}$ ($\mu$A/cm$^2$) | $E_{bd}$ (mV) | $i_{bd}$ ($\mu$A/cm$^2$) | Corrosion Rate (mm/Year) | Passive Range (mV) |
|---|---|---|---|---|---|---|
| CVC NiTi rod | $-334 \pm 4$ | $0.44 \pm 0.05$ | $329 \pm 4$ | $6.8 \pm 0.2$ | $(4.2 \pm 0.3) \times 10^{-3}$ | 430 |
| com. NiTi | $-300 \pm 4$ | $0.30 \pm 0.03$ | $634 \pm 7$ | $6.2 \pm 0.2$ | $(2.6 \pm 0.2) \times 10^{-3}$ | 730 |
| TiO$_2$/CVC NiTi rod | $-235 \pm 3$ | $0.34 \pm 0.03$ | $643 \pm 7$ | $6.2 \pm 0.2$ | $(3.1 \pm 0.2) \times 10^{-3}$ | 650 |
| TiO$_2$/com. NiTi | $-186 \pm 2$ | $0.16 \pm 0.02$ | / | / | $(1.1 \pm 0.1) \times 10^{-4}$ | / |

3.2.2. Electrochemical Impedance Spectroscopy

Electrochemical Impedance Spectroscopy (EIS) measurements were performed at Open-Circuit Potential conditions in simulated body fluid for 8 days. Figure 10 shows (for 4 selected examples: 12 h, 96 h, 168 h and 192 h) the Nyquist impedance diagrams for the CVC NiTi rod, commercial Nitinol, and also for both with a deposited TiO$_2$ layer at different times of immersion. The system response, shown through the Nyqvist plots, shows a typical depressed semicircle shape, and the response was increasing with the immersion time for all specimens.

The analysed data for the Nyquist plots predicted the equivalent circuits shown in Figure 11. For inhomogeneous layers, a similar equivalent circuit was applied by Izquierdo J. et al. [33] and Figueira N. et al. [34]. $R_1$ represents the resistance through the porous external oxide layer, while R$_2$ represents the resistance through the inner compact oxide layer. $R_S$ is the resistance of the solution, while CPE$_1$ and CPE$_2$ are constant phase elements corresponding to R$_1$ and R$_2$. The use of a Constant Phase Element (CPE) was required to confirm the non-ideal capacitive response observed as a depressed semicircle in the corresponding Nyquist diagrams. The CPE originates from the surface roughness and inhomogeneities present in the TiO$_2$ layers at the microscopic level. The equivalent circuit in Figure 9a has only resistance through the oxide layer and was used only for the specimens of CVC NiTi rod and commercial Nitinol in the first 48 h of immersion. After this exposure time, the equivalent circuit in Figure 9b was applied, and this equivalent circuit was also valid for all the immersion times for the TiO$_2$/CVC NiTi rod and TiO$_2$/commercial

Nitinol. The difference can be explained by not having an oxide layer at the beginning on the surface of the CVC NiTi rod and the commercial Nitinol. Therefore, for the first 48 h of immersion, the resulting layer is still very thin and compact, which can be represented by only 1 resistor. After this time, the oxide layer became thicker and inhomogeneous, so another resistance was added, which represents the resistance through the porous oxide layer. On the surface of the $TiO_2$/CVC NiTi rod and the $TiO_2$/commercial Nitinol specimens, a nanosized $TiO_2$ layer was deposited previously with ALD, so an equivalent circuit with two resistances was used from the beginning of the immersion.

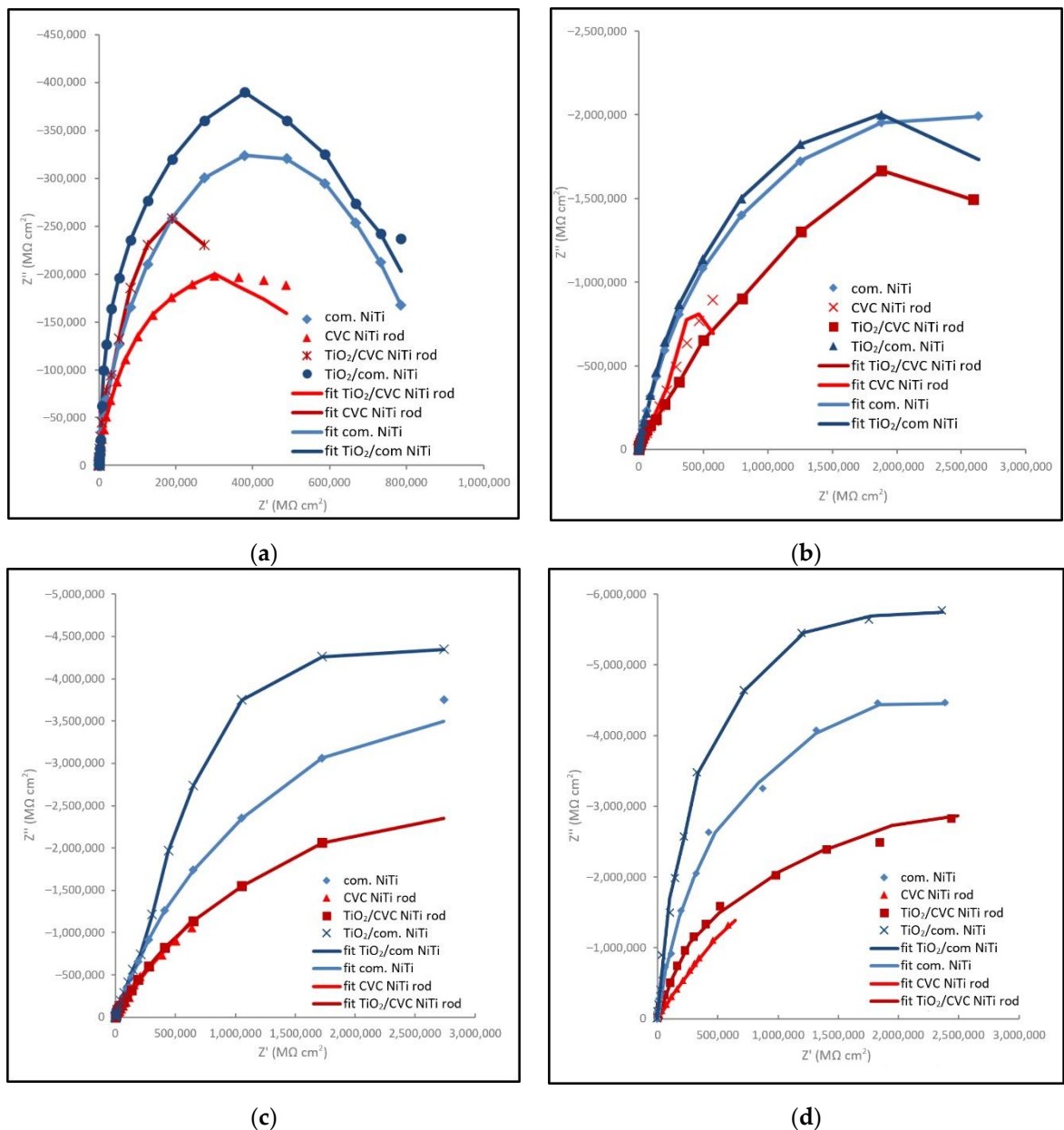

**Figure 10.** Nyquist diagrams for the CVC NiTi rod, commercial Nitinol, $TiO_2$/CVC NiTi rod and $TiO_2$/commercial Nitinol after (**a**) 12 h, (**b**) 96 h, (**c**) 168 h and (**d**) 192 h of immersion.

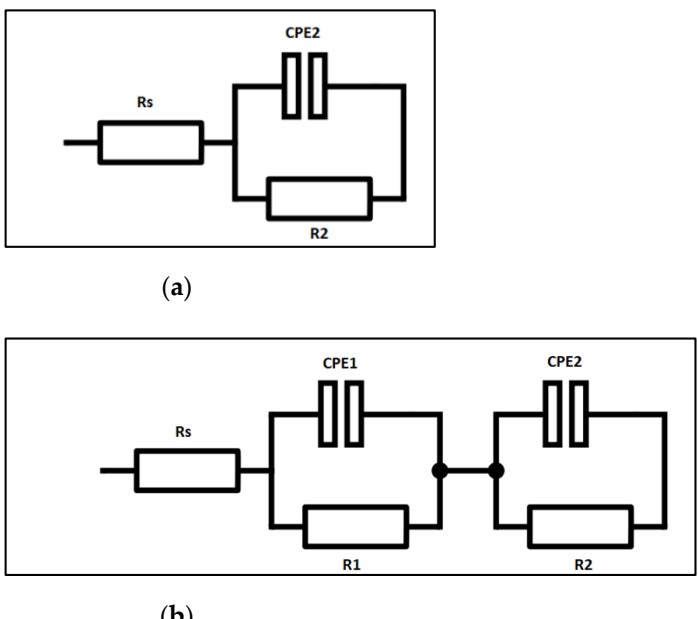

(a)

(b)

**Figure 11.** Equivalent circuits for the interpretation of the measured impedance spectra with: (**a**) one resistance and (**b**) two resistances.

Table 2 shows the resistance values of the porous layer ($R_1$) and the oxide layer ($R_2$) for all the specimens. The error obtained when fitting the EIS experimental data was below 0.02% for all the specimens. The CVC NiTi rod and the commercial Nitinol, for the first 48 h of immersion, had only resistance through the oxide layer as a consequence of the thin and compact layer ($R_2$). After this time, resistance through the porous layer was also considered. Both the $TiO_2$/CVC NiTi rod and $TiO_2$/commercial Nitinol had resistances through both layers for the total immersion time ($R_1 + R_2$), since the $TiO_2$ layer was previously deposited on the surface of the specimens. The resistance $R_1$ was considerably lower for all the specimens, which was not surprising since the solution will penetrate through the porous layer much more easily than through the compact layer. While the resistance through the oxide layer was increasing over the immersion time for all the specimens, the layer was getting thicker, so the resistance through the porous layer was quite constant.

**Table 2.** Porous corrosion resistance $R_1$ and oxide corrosion resistance $R_2$ of different specimens at certain times of immersion.

| t (h) | $R_{1com}$ $\times 10^5$ ($\Omega$) | $R_{2com}$ $\times 10^5$ ($\Omega$) | $R_{1CVC}$ $\times 10^5$ ($\Omega$) | $R_{2CVC}$ $\times 10^5$ ($\Omega$) | $R_{1TiO2/CVC}$ $\times 10^5$ ($\Omega$) | $R_{2TiO2/CVC}$ $\times 10^5$ ($\Omega$) | $R_{1TiO2/com}$ $\times 10^5$ ($\Omega$) | $R_{2TiO2/com}$ $\times 10^5$ ($\Omega$) |
|---|---|---|---|---|---|---|---|---|
| 1 | 0.00 | 3.74 | 0.00 | 2.08 | 0.24 | 2.22 | 0.82 | 4.00 |
| 6 | 0.00 | 6.86 | 0.00 | 2.30 | 0.42 | 3.29 | 0.94 | 8.20 |
| 12 | 0.00 | 6.49 | 0.00 | 2.61 | 0.79 | 3.85 | 1.16 | 10.20 |
| 24 | 0.00 | 7.41 | 0.00 | 2.71 | 1.36 | 4.41 | 1.34 | 18.01 |
| 48 | 0.00 | 10.94 | 0.00 | 2.55 | 2.44 | 7.58 | 1.45 | 25.00 |
| 72 | 4.32 | 19.65 | 0.33 | 7.80 | 2.59 | 13.80 | 1.37 | 33.81 |
| 96 | 6.92 | 40.35 | 0.28 | 18.00 | 2.48 | 22.51 | 1.56 | 52.51 |
| 120 | 7.46 | 56.88 | 0.34 | 24.08 | 2.66 | 33.00 | 1.35 | 68.50 |
| 144 | 8.32 | 70.29 | 0.33 | 28.06 | 2.68 | 44.72 | 1.34 | 89.00 |
| 168 | 9.01 | 89.47 | 0.35 | 30.25 | 2.69 | 56.51 | 1.26 | 101.04 |
| 192 | 10.28 | 81.47 | 0.31 | 29.62 | 2.29 | 60.31 | 1.46 | 113.56 |

Figure 12 represents the polarisation or the total corrosion resistance $R_p$ as a function of time. $R_p$ can be calculated according to equation:

$$R_p = R_1 + R_2, \tag{3}$$

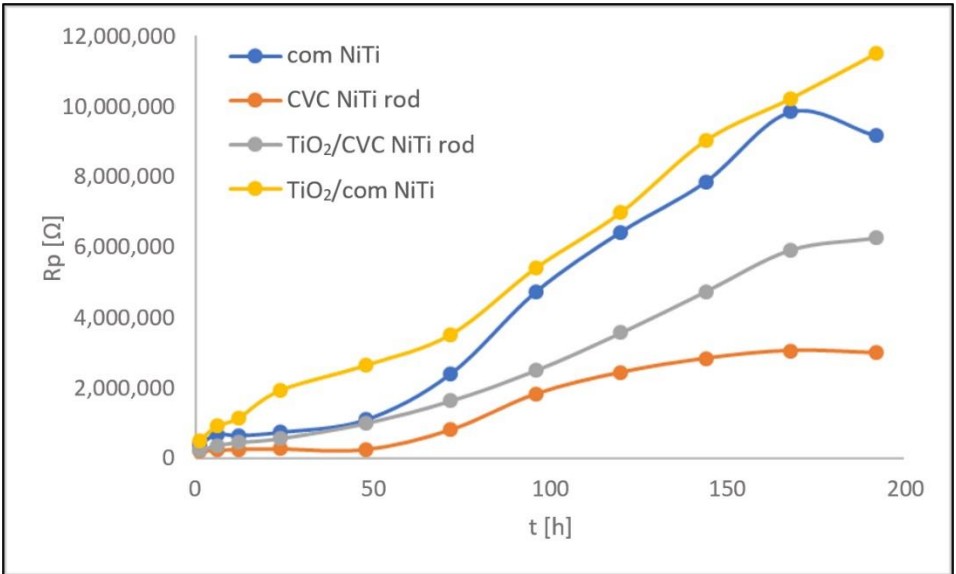

**Figure 12.** Total corrosion resistance vs time of exposure for all tested specimens.

The diagram of total corrosion resistance (Figure 12) shows clearly that the corrosion resistance increased with time for all 4 specimens. This was a consequence of the formation of a protective oxide layer. In the case when the $TiO_2$ layer was deposited on the specimen, it is noticeable that the corrosion resistance was increasing in the initial time, while for the CVC NiTi rod and commercial Nitinol, in the initial 48 h, there was no significant change in resistance as a consequence of the thin oxide layer. The highest corrosion resistance was for the $TiO_2$/commercial Nitinol, followed by the commercial Nitinol and the $TiO_2$/CVC NiTi rod, and the poorest corrosion resistance was for the CVC NiTi rod. The cause of the poorer corrosion properties of the CVC NiTi rod was inhomogeneity in the chemical composition and higher nickel content than in the commercial NiTi.

## 4. Conclusions

The following conclusions can be drawn from the current research work:

1.  The thickness of the formatted $TiO_2$ layers on the CVC NiTi rod was 52.2 nm, and on the commercial Nitinol it was 51.7 nm.
2.  The formatted $TiO_2$ layers were confirmed by XPS and SEM/EDX analyses.
3.  The high-energy resolution XPS spectra for $TiO_2$ from both specimens were very similar. The Ti $2p_{3/2}$ peak at 458.6 eV and the Ti $2p_{1/2}$ peak at 464.5 eV were observed on the surface of both specimens. This corresponds to the binding energy, which is related with the Ti(4+) oxidation state. This shows the presence of the $TiO_2$ compound on the surfaces of both investigated ALD $TiO_2$-covered specimens.
4.  The potentiodynamic test showed that the passive layer on the $TiO_2$/commercial Nitinol was the most stable and resistant to external corrosion influences. The corrosion stability fell from the commercial Nitinol and the CVC NiTi rod with and without the $TiO_2$ layer.
5.  The corrosion rate was the smallest for $TiO_2$/commercial Nitinol; the $TiO_2$/CVC NiTi rod and commercial Nitinol were far away but very close together, while the CVC NiTi rod showed the highest corrosion rate, or the lowest corrosion resistance, by the potentiodynamic test.
6.  Electrochemical Impedance Spectroscopy is interpreted using the Nyquist impedance diagrams, where a typical depressed semicircle shape and the response increasing with the immersion time were shown for all specimens.
7.  With the help of using the resistance through the porous external oxide layer ($R_1$) and the resistance through the inner compact oxide layer ($R_2$), it was determined that

the CVC NiTi rod and the commercial Nitinol had, for the first 48 h of immersion, only resistance through the oxide layer as a consequence of the thin and compact layer ($R_2$). On the other hand, the $TiO_2$/CVC NiTi rod and $TiO_2$/commercial Nitinol had resistances through both layers for the total immersion time ($R_1 + R_2$). The resistance $R_1$ was considerably lower for all the specimens, which was not surprising since the solution will penetrate through the porous layer much more easily than through the compact layer.

8.  It was proven that adding a $TiO_2$ layer on the Nitinol surface was significant for improving the corrosion resistance, but a decisive role can be still attributed to the chemical composition and microstructure of the substrate that the ALD coating is applied.

**Author Contributions:** Conceptualization, R.R., A.S. and A.K.; methodology, R.R., A.S. and A.K.; validation, R.R. and A.S.; formal analysis, A.S.; investigation, A.S..; resources, R.R. and A.K.; writing—original draft preparation, R.R. and A.S.; writing—review and editing, R.R.; visualisation, A.S.; supervision, R.R.; funding acquisition, R.R. and A.K. All authors have read and agreed to the published version of the manuscript.

**Funding:** This research was funded by the Slovenian Research Agency—Applied Project No.: L2-5486 and the young researchers programme—the PhD grant no. 10/2013.

**Data Availability Statement:** Not Applicable.

**Acknowledgments:** Special thanks go to Janez Kovač from the Jožef Stefan Institute Ljubljana Slovenia for performing the XPS analysis.

**Conflicts of Interest:** The authors declare no conflict of interest.

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
