# Peer review of "Atomic Layer Deposition of aTiO2 Layer on Nitinol and Its Corrosion Resistance in a Simulated Body Fluid"

_metals, doi:10.3390/met11040659_

Round 1

Reviewer 1 Report

The submission is devoted to studying corrosion properties of the atomic layer deposited TiOx coatings on nitinol. The topic of the research is really important and relevant. Indeed the nitinol has many advantages over other biomedical alloys. But the danger of nickel dissolution is very high and this limits the use of nitinol in medicine. The motivation of the authors and the structure of the study is very clear. The results are promising, but the manuscript has many drawbacks.

The authors have prepared a very good and logical introduction that shows why additional coating is needed for nitinol, but they have provided very few specific references. For example, the phrase: "Ni releasing can induce toxic, allergic and hypersensitive reactions or tissue necrosis after long term implantation" must be confirmed with references. "The low-temperature (60-160 °C) pre-treatment protocols or high-temperature annealing in the air used for deposition of a thick TiO2 layer onto the Nitinol surface results in Ni accumulation in the surface depth." it is also necessary to confirm it with references.

Composition and structure are poorly studied. There is no XRD study. Although the crystal structure can have a significant effect on the corrosion resistance of coatings. As XPS results it was shown only survey spectra. The peaks shown in the survey spectrum are not labeled…). Is there metallic Ti or Ni? Is the ALD coating continuous and conformal? Moreover, survey spectra cannot show detailed chemical composition. From these spectra, it is impossible to draw conclusions about which titanium oxide was obtained (TiO2, TiO, TiOx…).

The authors stated: «Figure 4 shows a typical XPS analysis of TiO2.» It is not a typical analysis. For the typical analysis authors should present the Ti2p, O1s, C1s… spectra and deconvolute them into separate components if it is possible (For example TiO2, TiO, Ti for Ti2p spectra). Unfortunately, the presented data and spectrum do not allow us to draw any conclusions about the composition of the coating.

The authors prepared the “made in house CVC NiTi”. Unfortunately, there is no description of the procedure for NiTi preparation or references to a description.

In section 2.1 authors stated: «TiCl4 is adsorbed to the sample’s surface and reacts there with reactive sites (initially these are organic impurities).» What are the organic impurities? Probably the authors did not fully understand the ALD mechanism. TiCl4 reacts with hydroxyl groups or other surface species but no impurities!

How many ALD cycles authors use? It is necessary to indicate this in the Materials and Methods. Also, there is no information about the SEM study in the Materials and Methods.

The manuscript also has many typos and inaccuracies in the presentation of results. For example, in Fig. 8 shows two absolutely identically marked dependencies (red circles).

The authors are not the first who protected nitinol with coatings. In addition, ALD has already been used to produce the anticorrosion coating for nitinol. A brief literature search shows that TiO2 (https://doi.org/10.1007/s11665-018-3136-x) and Al2O3 (doi:10.3390/coatings10080746) ALD coatings were deposited to protect nitinol. But the authors do not compare their data with the literature. Therefore, unfortunately, it is impossible to assess the importance and impact of the work without such a comparison.

Abstract and conclusion should be revised. The abstract contains only a brief introduction to the topic of the study and a description of the experiments but there are no results and conclusions. The conclusion section also contains few concrete results. Conclusion #1 is very obvious and conclusion #4 is not confirmed by the results of the research.

Author Response

Dear

Thanks for the review, which we read carefully. Below we send the answers and the necessary additions/improvements. The text has been reviewed again by the native speaker Shelagh Hedges. We think now, hopefully, that the article is now suitable for publication in Metals. On behalf of all the authors, I am sending best regards.

Rebeka Rudolf

The submission is devoted to studying corrosion properties of the atomic layer deposited TiOx coatings on nitinol. The topic of the research is really important and relevant. Indeed the nitinol has many advantages over other biomedical alloys. But the danger of nickel dissolution is very high and this limits the use of nitinol in medicine. The motivation of the authors and the structure of the study is very clear. The results are promising, but the manuscript has many drawbacks.

The authors have prepared a very good and logical introduction that shows why additional coating is needed for nitinol, but they have provided very few specific references. For example, the phrase: "Ni releasing can induce toxic, allergic and hypersensitive reactions or tissue necrosis after long term implantation" must be confirmed with references. "The low-temperature (60-160 °C) pre-treatment protocols or high-temperature annealing in the air used for deposition of a thick TiO2 layer onto the Nitinol surface results in Ni accumulation in the surface depth." it is also necessary to confirm it with references.

We have added references – for the phrase "Ni releasing can induce toxic, allergic and hypersensitive reactions or tissue necrosis after long term implantation"

(8) Wawrzynski, J.; Gil, J.A.; Goodman A.D.; Waryasz G.R. Hypersensitivity to Orthopedic Implants: A Review of the Literature. Rheumatology and Therapy. 2017, 4(1), 45-56.

(9) Teo, Z.W.W.; Schalock, P.C. Hypersensitivity Reactions to Implanted Metal Devices: Facts and Fictions. J Investig Allergol Clin Immunol. 2016; 26(5), 279-29.

(10) Nordberg. G.F., Gerhardsson, L.; Broberg, K.; Mumtaz, M.; Ruis, P.; Fowler, B.A. Interactions in Metal Toxicology. Handbook on the Toxicology of Metals (Third Edition). 2007, 117-145.

And references for: The low-temperature (60-160 °C) pre-treatment protocols or high-temperature annealing in the air used for deposition of a thick TiO2 layer onto the Nitinol surface results in Ni accumulation in the surface depth”

(11) Shabalovskaya, S.; Anderegg, J.; Humbeeck, J.V. Critical overview of Nitinol surfaces and their modifications for medical applications, Acta Biomaterialia. 2008, 4, 447–467.

(12) Pohl, M.; Glogowski, T.; Kuehn, S.; Hessing, C.; Unterumsberger, F. Formation of titanium oxide coatings on NiTi shape memory alloys by selective oxidation. Materials Science and Engineering A. 2008, 481–482(1), 123–126.

(13) Nasakina, E.O.; Sudarchikova, M.A.; Sergienko, K.V.; Konushkin, S.V.; Sevost’yanov, M.A. Ion Release and Surface Characterization of Nanostructured Nitinol during Long-Term Testing. Nanomaterials. 2019, 9, 1569, 2-25.

Composition and structure are poorly studied. There is no XRD study. Although the crystal structure can have a significant effect on the corrosion resistance of coatings. As XPS results it was shown only survey spectra. The peaks shown in the survey spectrum are not labeled…). Is there metallic Ti or Ni? Is the ALD coating continuous and conformal? Moreover, survey spectra cannot show detailed chemical composition. From these spectra, it is impossible to draw conclusions about which titanium oxide was obtained (TiO2, TiO, TiOx…).

In this study XRD analysis was not made due to the extremely thin layers (nm). XPS is a more appropriate analytical method for very thin nano-coating, which was carried out. We have shown one of the characteristic XPS spectrums for the formatted ALD layer (Fig. 5). In this Figure the mark for the Ti2p peak has been added to the diagram – as required. This is a peak for metallic Ti. The obtained XPS spectrum is typical for TiO2 (with characteristic shape and position) - based on XPS spectra from the database of the SPECS spectrometer. The calculation for TiO2 was made from the height of the peaks, 1x of titanium and 2x the height of oxygen, where the ratio should follow 1:2.

The ALD formed TiO2 layer is continuous.

On the thin layer it is very difficult to perform chemical analysis in order to confirm the content of the elements. With a technique like SEM/EDX it is not possible to identify chemical composition, as the analysis includes a significantly larger volume (> 3-5 µm3) than the thickness of the layer itself.

We performed additional line EDX analysis using the Jeol JSM-7800F Field Emission Scanning Electron Microscope, see new added text: 2. Materials and Methods (lines: 156-172), 3. Results and Discussion (lines 223-254) and Figure 6.

The authors stated: «Figure 4 shows a typical XPS analysis of TiO2.» It is not a typical analysis. For the typical analysis authors should present the Ti2p, O1s, C1s… spectra and deconvolute them into separate components if it is possible (For example TiO2, TiO, Ti for Ti2p spectra). Unfortunately, the presented data and spectrum do not allow us to draw any conclusions about the composition of the coating.

A Ti2p peak was added to the diagram and a new explanation, as set out in the point above.

The authors prepared the “made in house CVC NiTi”. Unfortunately, there is no description of the procedure for NiTi preparation or references to a description.

A reference has been added:

(29) Lojen, G.; Stambolić, A.; Šetina, B.; Rudolf, R. Experimental continuous casting of nitinol. Metals. 2020, 10(4), 1-14.

In section 2.1 authors stated: «TiCl4 is adsorbed to the sample’s surface and reacts there with reactive sites (initially these are organic impurities).» What are the organic impurities? Probably the authors did not fully understand the ALD mechanism. TiCl4 reacts with hydroxyl groups or other surface species but no impurities!

In ideal circumstances the surface of nitinol is titanium and nickel, but there are also some other factors, like human or environment, with which we add some hydrocarbon impurities to the surface. These hydrocarbons can have reactive sites, like hydroxyl, carboxyl, nitrile groups, with which Ti can react in the first cycle.

How many ALD cycles authors use? It is necessary to indicate this in the Materials and Methods. Also, there is no information about the SEM study in the Materials and Methods.

The information was listed in the Results Section and was added to the Materials Section as well. The results of the SEM/EDX line scan analysis have been added.

The manuscript also has many typos and inaccuracies in the presentation of results. For example, in Fig. 8 shows two absolutely identically marked dependencies (red circles).

In Figure 8 (now Figure 10) the colour of the lines has been changed to make them more clearly visible.

The authors are not the first who protected nitinol with coatings. In addition, ALD has already been used to produce the anticorrosion coating for nitinol. A brief literature search shows that TiO2 (https://doi.org/10.1007/s11665-018-3136-x) and Al2O3 (doi:10.3390/coatings10080746) ALD coatings were deposited to protect nitinol. But the authors do not compare their data with the literature. Therefore, unfortunately, it is impossible to assess the importance and impact of the work without such a comparison.

We have added both the above mentioned references. Moreover, we have explained what was done before and what is new in our research (study) - impact. Please see lines: 102-108.

(27) Vokoun, D.; Racek, J.; Kaderavek, L.; Kei, C.C.; Yu, Y.S.; Klimša, L.; Šittner, P. Atomic Layer-Deposited TiO2Coatings on NiTi Surface. JMEPEG. 2018, 27:572–579.

(28) Vokoun, D.; Klimša, L.; Vetushka, A.; Duchoň, J.; Racek, J.; Drahokoupil, J.; Kopeček, J.; Yu, Y.S.; Koothan; N.; Kei, C.C. Al2O3 and Pt Atomic Layer Deposition for Surface Modification of NiTi Shape Memory Films. Coatings. 2020, 10(8):746.

Abstract and conclusion should be revised. The abstract contains only a brief introduction to the topic of the study and a description of the experiments but there are no results and conclusions. The conclusion section also contains few concrete results. Conclusion #1 is very obvious and conclusion #4 is not confirmed by the results of the research.

Both Abstract and Conclusions have been improved.

Reviewer 2 Report

The problem in the paper by Rudolf et al. is quite clearly stated, as related to the reduction of the harmful influence of surface nickel in Nitinol by coating it with a corrosion resistant and, at the same time, biocompatible TiO2 film. TiO2 film is also quite a natural choice, because it is – presumably – also compatible to titanium-containing alloys such as Nitinol, in terms of adhesion and stability.

The main focus of the paper seems to lie on the proof of corrosion resistance comparatively measured for TiO2-coated and non-coated reference Nitinol substrates. This seems to be provided (Fig. 8).

There were few questions arisen, connected mainly to the presentation quality. Since that may affect the credibility of the scientific paper, it is highly recommended to address these minor issues described below as follows.

  1. It is to be noted that the title of the manuscript is wrongly written. The paper is, namely, not about atomic layer deposition of nitinol, but atomic layer deposition of titanium oxide corrosion protection layer on nitinol. Nitinol itself as an alloy of titanium and nickel metals, was not deposited by ALD in this paper.
  2. It is written in the sentence in the row 87, that „The most common methods employed are anodisation, plasma spraying, Atomic Layer Deposition (ALD), etc.“ It can not be understood for what these methods are „common.“ In addition, it seems that there is quite a list considered, et cetera. So, what are then the most common methods for which applications?
  3. Between rows 167 and The scanning measurement results of XPS are not diagrams, these are called spectra. In the caption to Fig. 4, it is written correctly, although such general spectra should be called survey spectra, they do not yet visually characterize the details of the binding or species recognized.
  4. The XPS results are loosely presented. At first it is not understood, what exactly is meant by the sentence in the rows 170-171, that „Because the ratio of intensities between 1Ti and 2O is about 1:2,“ the deposited layer on the surface of the samples is actually TiO2.“ What are the meanings of „about“ and „between 1Ti and 2O“ here? And which intensities are taken into account, actually? Were these the maximum peaks intensities only, or where the peak areas, absorption cross-sections of the elements and peak widths also considered in the estimations? It seems that the XPS analysis has been heavily oversimplified in this paper.
  5. The TiO2 films have, in the present work, grown to the thicknesses exceeding 50 nm. Instead of the XPS, for which the information depth does not exceed 2-3 nm, and which requires rather complex approach in terms of the analysis of results, the authors could have applied much more convenient X-ray diffractometry. Namely, TiO2 films, as thick as 50 nm, deposited from titanium chloride even at as low as 250 degrees, should easily grow polycrystalline on any substrate. Thus, by means of XRD, the formation of titanium dioxide should be recognized conveniently and throughout the film thickness, which is most adequate keeping in mind its tasks as corrosion-resistant coating. Indeed, why did the authors not apply XRD as an analysis tool.

Author Response

Dear!

Thanks for the review, which we read carefully. Below we send the answers and the necessary additions/improvements. The text has been reviewed again by the native speaker Shelagh Hedges. We think now, hopefully, that the article is now suitable for publication in Metals. On behalf of all the authors, I am sending best regards.

Rebeka Rudolf

The problem in the paper by Rudolf et al. is quite clearly stated, as related to the reduction of the harmful influence of surface nickel in Nitinol by coating it with a corrosion resistant and, at the same time, biocompatible TiO2 film. TiO2 film is also quite a natural choice, because it is – presumably – also compatible to titanium-containing alloys such as Nitinol, in terms of adhesion and stability.

The main focus of the paper seems to lie on the proof of corrosion resistance comparatively measured for TiO2-coated and non-coated reference Nitinol substrates. This seems to be provided (Fig. 8).

There were few questions arisen, connected mainly to the presentation quality. Since that may affect the credibility of the scientific paper, it is highly recommended to address these minor issues described below as follows.

  1. It is to be noted that the title of the manuscript is wrongly written. The paper is, namely, not about atomic layer deposition of nitinol, but atomic layer deposition of titanium oxide corrosion protection layer on nitinol. Nitinol itself as an alloy of titanium and nickel metals, was not deposited by ALD in this paper.

Thank you for your comment. The title has been changed accordingly: “Atomic layer deposition of TiO2 layer on Nitinol and its corrosion resistance in a simulated body fluid”.

  1. It is written in the sentence in the row 87, that „The most common methods employed are anodisation, plasma spraying, Atomic Layer Deposition (ALD), etc.“ It cannot be understood for what these methods are „common.“ In addition, it seems that there is quite a list considered, et cetera. So, what are then the most common methods for which applications?

Those are the common methods for layer formation and were added to the original test.

  1. Between rows 167 and The scanning measurement results of XPS are not diagrams, these are called spectra. In the caption to Fig. 4, it is written correctly, although such general spectra should be called survey spectra, they do not yet visually characterize the details of the binding or species recognized.

Thank you for your comment. The text has been improved (see lines: 214-220).

  1. The XPS results are loosely presented. At first it is not understood, what exactly is meant by the sentence in the rows 170-171, that „Because the ratio of intensities between 1Ti and 2O is about 1:2,“ the deposited layer on the surface of the samples is actually TiO2.“ What are the meanings of „about“ and „between 1Ti and 2O“ here? And which intensities are taken into account, actually? Were these the maximum peaks intensities only, or where the peak areas, absorption cross-sections of the elements and peak widths also considered in the estimations? It seems that the XPS analysis has been heavily oversimplified in this paper.

We agree with your comment. The text of the XPS results` explanation has been improved and corrected (see lines: 214-254).

  1. The TiO2 films have, in the present work, grown to the thicknesses exceeding 50 nm. Instead of the XPS, for which the information depth does not exceed 2-3 nm, and which requires rather complex approach in terms of the analysis of results, the authors could have applied much more convenient X-ray diffractometry. Namely, TiO2 films, as thick as 50 nm, deposited from titanium chloride even at as low as 250 degrees, should easily grow polycrystalline on any substrate. Thus, by means of XRD, the formation of titanium dioxide should be recognized conveniently and throughout the film thickness, which is most adequate keeping in mind its tasks as corrosion-resistant coating. Indeed, why did the authors not apply XRD as an analysis tool.

We are grateful for this comment and the research proposal (XRD), which we have not and cannot do now. However, we will take this into account in future research activities. We have added the SEM/EDX performed results of the resulting TiO2 layer.

Reviewer 3 Report

The atomic layer deposition (ALD) technique for producing TiO2 on nitinol was conducted in the present investigation. Then, the corrosion behavior of the deposited TiO2 layers on the different matrices was tested in a simulated body fluid. It is an interesting issue for nitinol application. The manuscript should be major revised before accepting for publication.

The authors should consider the following problems:

(1) The peaks corresponding to Ti element should be noted in Fig.4.

(2) The comments are not convinced enough on page 5, line 170-171: Because the ratio of intensities between 1Ti and 2O is about 1: 2, the deposited layer on the surface of the samples is actually TiO2. Other evidence should be provided.

(3) The qualities of the figures should be improved, especially for Fig.6 and Fig.7.

(4) The potentiodynamic curves of the CVC NiTi rod and the commercial NiTi alloy were already reported in the authors’ previous paper (reference [21]). Some relevant contents could be cited, instead of putting in the present manuscript again. In addition, there are little big differences in the data between previous table 1 and present table 1, also for table 2, Why?

(5) On page 8, line 233: CVT should be CVC.

Author Response

Dear!

Thanks for the review, which we read carefully. Below we send the answers and the necessary additions/improvements. The text has been reviewed again by the native speaker Shelagh Hedges. We think now, hopefully, that the article is now suitable for publication in Metals. On behalf of all the authors, I am sending best regards.

Rebeka Rudolf

The atomic layer deposition (ALD) technique for producing TiO2 on nitinol was conducted in the present investigation. Then, the corrosion behavior of the deposited TiO2 layers on the different matrices was tested in a simulated body fluid. It is an interesting issue for nitinol application. The manuscript should be major revised before accepting for publication.

The authors should consider the following problems:

(1) The peaks corresponding to Ti element should be noted in Fig.4.

We agree with your comment. Figure 4 (now Fig. 5) has been improved and corrected.

(2) The comments are not convinced enough on page 5, line 170-171: Because the ratio of intensities between 1Ti and 2O is about 1: 2, the deposited layer on the surface of the samples is actually TiO2. Other evidence should be provided.

We agree with your comment. The text of XPS results explanation has been improved and corrected (see lines: 214-255).

(3) The qualities of the figures should be improved, especially for Fig.6 and Fig.7.

We agree with your comment. The figures has been improved.

(4) The potentiodynamic curves of the CVC NiTi rod and the commercial NiTi alloy were already reported in the authors’ previous paper (reference [21]). Some relevant contents could be cited, instead of putting in the present manuscript again. In addition, there are little big differences in the data between previous table 1 and present table 1, also for table 2, Why?

In reference (Stambolić, A.; Jenko, M.; Kocijan, A.; Žužek B.; Drobne, D.; Rudolf, R. Determination of mechanical and functional properties by continuous vertical cast NiTi rod. Materiali in tehnologije 2018, 52, 5, 521-527) CVC NiTi rod of another experiment was used for analysis. Therefore, there was a difference in the data in the tables (present, previous). It should be mentioned here that in prevously paper, this was the first and only attempt of analyzing the data until that paper was written. Namely such study neeed a lot of practice to fully understand the software for analyzing the data:  extrapolation of potentiodynamic curve can cause human errors and by EIS you need to guess the resistances. Based on this the subject of present scientific is discussion, where a lot more corrosion analysis were made (3 for each sample) and a lot of work to evaluate the data was done after that.

(5) On page 8, line 233: CVT should be CVC.

Improved.

Reviewer 4 Report

The manuscript entitled „Atomic layer deposition of Nitinol and its corrosion resistance in a simulated body fluid” is considered to be relevant to the scope of this journal.

However, several points need to be addressed prior to publication of this manuscript. My comments/suggestions are given:

  1. The statement in the paragraph between the lines 119-121 “The chemical composition of the CVC NiTi rod varied through the cross and longitudinal sections, because the drawing process was not optimal.” is not supported. To prove this statement, an in-depth mapping SEM or EDS profile must be added.
  2. The authors refer throughout the manuscript to the "50-nm TiO2 layer". Since Figures 2a and 2b show that the layers are about 50 nm, I think this expression should be corrected.
  3. The difference between Figures 2a and 2b cannot be made very easily. The figures must be spaced apart. The figures should be replaced with others with a higher resolution.
  4. Line 181 “Corrosion potential and current, and breakdown potential and current values were obtained by graphic extrapolation.” It is not understood. It needs to be reformulated.
  5. In Figures 5a and 5b, the logarithmic representation is not marked correctly. The writing mode must be changed.
  6. In order to have a comparison of the samples, Figure 5 should be reorganized. Either all the curves will be presented in a single graph, or they will be compared in Figure 5a CVC NiTi with and without TiO2 and Figure 5b com. NiTi with and without TiO2. The authors must verify the legend of the figures. According to the polarization curves behavior and according to the data presented in Table 1, I think it is something wrong.
  7. Table 1 shows the results for the corrosion current (Icorr) or for the corrosion current density? Since the corrosion current density is represented in Figure 5, I think that the same should appear in the Table, especially since the surface exposed to corrosion tests is not specified in the Materials and Methods chapter.
  8. The corrosion rate in Table 1 is given with the unit of measurement (mmpy). Please be more precise.
  9. The data contained in the paragraph between lines 188-192 “The CVC NiTi rod had the smallest passive range (from -100 mV to 330 mV, which is 430 mV), followed by TiO2/CVC NiTi rod (650 mV), com. NiTi (730 mV) and TiO2/com. NiTi. CVC NiTi rod (329.2 mV) had the lowest breakdown potential, followed by com. NiTi (633.8 mV) and TiO2/CVC NiTi rod (642.5 mV), while TiO2/com”can be aded as a new column in Table 1.
  10. The authors must also make a comparison of the results obtained by them with the existing data in the literature.
  11. The number of samples analyzed in each experiment must be specified.
  12. The errors for the values obtained in Table 1 must be specified.
  13. You can explain the paragraph “The corrosion potential was the highest for TiO2 / com. NiTi (-185.9 mV), which means that the passive layer on this sample was the most stable and resistant to external influences. ”? Corrosion resistance can be assessed based on the value of the corrosion current density rather…
  14. If EIS measurements were performed for 8 days, why are the results presented only up to 168 hours?
  15. Table 2 must be organized differently because it is very difficult to follow. Also, the values obtained for the other elements from the equivalent circuit must be presented. The errors obtained when fitting the experimental data must be entered because only in this way we can know if the proposed equivalent circuits are correct or not.

Author Response

Dear!

Thanks for the review, which we read carefully. Below we send the answers and the necessary additions/improvements. The text has been reviewed again by the native speaker Shelagh Hedges. We think now, hopefully, that the article is now suitable for publication in Metals. On behalf of all the authors, I am sending best regards.

Rebeka Rudolf

The manuscript entitled „Atomic layer deposition of Nitinol and its corrosion resistance in a simulated body fluid” is considered to be relevant to the scope of this journal. However, several points need to be addressed prior to publication of this manuscript. My comments/suggestions are given: 

  1. The statement in the paragraph between the lines 119-121 “The chemical composition of the CVC NiTi rod varied through the cross and longitudinal sections, because the drawing process was not optimal.” is not supported. To prove this statement, an in-depth mapping SEM or EDS profile must be added.

We agree with your comment. The text has been improved and new Figure 2 has been added.

  1. The authors refer throughout the manuscript to the "50-nm TiO2 layer". Since Figures 2a and 2b show that the layers are about 50 nm, I think this expression should be corrected.

The text has been corrected with more precise indications of the dimensions of the resulting layer.

  1. The difference between Figures 2a and 2b cannot be made very easily. The figures must be spaced apart. The figures should be replaced with others with a higher resolution.

               Thank you for your valuable comment. We have spaced both Figures 2a and 2b apart for better           visualisation. We would also like to point out that these images were made at magnification    200.000X and could not reach a better resolution.

  1. Line 181 “Corrosion potential and current, and breakdown potential and current values were obtained by graphic extrapolation.” It is not understood. It needs to be reformulated.

               Thank you for the comment. We have rephrased this part of the manuscript accordingly:                “Corrosion potentials (Ecorr) and corrosion current densities (icorr) were obtained from the Tafel             region. Following that region, the specimens exhibited a passive region, which was limited by   the breakdown potential (Ebd), corresponding to the transpassive oxidation of the metal            species.”

  1. In Figures 5a and 5b, the logarithmic representation is not marked correctly. The writing mode must be changed.

               Thank you for your comment, the logarithmic representation has been changed now in                Figure 7.

  1. In order to have a comparison of the samples, Figure 5 should be reorganized. Either all the curves will be presented in a single graph, or they will be compared in Figure 5a CVC NiTi with and without TiO2 and Figure 5b com. NiTi with and without TiO2. The authors must verify the legend of the figures. According to the polarization curves behavior and according to the data presented in Table 1, I think it is something wrong.

The corrosion test and analyses were done 3 years ago, and Dr. Stambolič is not employed at the Institute anymore. Because of a researcher`s mistake all the data were lost. Otherwise, there is nothing wrong with the data in the Table and curves - they are correct.

  1. Table 1 shows the results for the corrosion current (Icorr) or for the corrosion current density? Since the corrosion current density is represented in Figure 5, I think that the same should appear in the Table, especially since the surface exposed to corrosion tests is not specified in the Materials and Methods chapter.

               Thank you. We have changed the corrosion current to the corrosion current density in Table     1, and also added the exposed surface to the Materials and Methods chapter.

  1. The corrosion rate in Table 1 is given with the unit of measurement (mmpy). Please be more precise.

               Thank you. We have changed the unit to mm/year.

  1. The data contained in the paragraph between lines 188-192 “The CVC NiTi rod had the smallest passive range (from -100 mV to 330 mV, which is 430 mV), followed by TiO2/CVC NiTi rod (650 mV), com. NiTi (730 mV) and TiO2/com. NiTi. CVC NiTi rod (329.2 mV) had the lowest breakdown potential, followed by com. NiTi (633.8 mV) and TiO2/CVC NiTi rod (642.5 mV), while TiO2/com”can be aded as a new column in Table 1.

               Thank you for your valuable comment. We have added these data to Table 1.

  1. The authors must also make a comparison of the results obtained by them with the existing data in the literature.

New text has been added in the Introduction explaining what was done before and what is new in our research (study) - impact. Please see lines: 102-108.

We have added two new references:

(27) Vokoun, D.; Racek, J.; Kaderavek, L.; Kei, C.C.; Yu, Y.S.; Klimša, L.; Šittner, P. Atomic Layer-Deposited TiO2Coatings on NiTi Surface. JMEPEG. 2018, 27:572–579.

(28) Vokoun, D.; Klimša, L.; Vetushka, A.; Duchoň, J.; Racek, J.; Drahokoupil, J.; Kopeček, J.; Yu, Y.S.; Koothan; N.; Kei, C.C. Al2O3 and Pt Atomic Layer Deposition for Surface Modification of NiTi Shape Memory Films. Coatings. 2020, 10(8):746.

  1. The number of samples analyzed in each experiment must be specified.

       3 samples were analysed – text has been added to the Materials section, Part 2.

  1. The errors for the values obtained in Table 1 must be specified.

               Thank you for your comment. We added the errors in Table 1.

  1. You can explain the paragraph “The corrosion potential was the highest for TiO2 / com. NiTi (-185.9 mV), which means that the passive layer on this sample was the most stable and resistant to external influences. ”? Corrosion resistance can be assessed based on the value of the corrosion current density rather…

               Thank you for your comment. We have changed the text as suggested: The corrosion current                density is the lowest for TiO2/com. NiTi (0.16 μA/cm2), which means that the passive layer on               this sample is the most stable and resistant to external influences.

  1. If EIS measurements were performed for 8 days, why are the results presented only up to 168 hours?

       The results in the Table are presented for 192 h, but there was some mistake in Figure 10   (now), which has been corrected (we have corrected the curves so they include data for 192               hours -8 days).

  1. Table 2 must be organized differently because it is very difficult to follow. Also, the values obtained for the other elements from the equivalent circuit must be presented. The errors obtained when fitting the experimental data must be entered because only in this way we can know if the proposed equivalent circuits are correct or not.

We consider that Table 2 is properly prepared, as all the data for the testing hours (1,6,12,… 192), separately for each hour, are on one line. We don’t think a different scheme would be better for readers.

Round 2

Reviewer 1 Report

Thanks to the authors for revising the article. But they did not answer all the questions. I still have a lot of questions and comments.

The main problems of the article are related to the low level of characterization of the composition and structure of the obtained coatings!!! By the way, reviewer №2 has essentially the same claims.

The authors state that the TiOx films are too thin for XRD. I have a lot of experience in studying titanium oxide films prepared by ALD. In some cases, it is possible to obtain characteristic XRD patterns even for films 10 nm thick. The authors have grown films with a thickness of 50nm. Therefore, I cannot accept such an answer from the authors. The only real problem with XRD measurements is the roughness of the surface (substrate). Therefore, the surface must be well polished before measurement, or polished witnesses must be used.

But the lack of XRD data is not the main disadvantage of the manuscript. The main problem is XPS !!!

  • Authors stated in the revision version of the manuscript that: « If we take this into account, the indicative ratio of peaks intensities between Ti and O, which is around 1:2, indicates that TiO2 was formed indirectly.»

This is totally wrong! Anyone can notice that this ratio is very far from 2/1, but rather closer to 1/1. But this, in fact, is absolutely irrelevant. The intensities in the XPS do not show anything important. Quantitative analysis and ratios of elements can only be obtained from the ratios of areas under the XPS peaks. But even so, you need to take into account the special coefficients that are different for different elements and instruments!!!

  • The authors stated in the revision version of the manuscript: «In this spectrum there are two main peaks: Ti2p at a binding energy of 469.2 eV and O1s at a binding energy of 530.7 eV.» This cannot be seen from the presented figure 5. Why 469.2 and not 469.9! Or even 490!!! The scale of the survey spectrum is such that it is absolutely impossible to distinguish it. I have already asked the authors to show individual spectra of Ti2p, O1s, C1s levels, but my request was ignored. Also, what was the measurement interval? 0.1 eV? I believe that the step (interval) was much larger.
  • Also, if you are present the spectrum in a scientific paper, then all peaks must be identified and signed. Unfortunately, the authors only signed Ti2p and O1s. For example, I can clearly see the carbon peak at 285 eV. But the authors do not mention it
  • In addition, the spectrum shown in figure 5 is not an XPS spectrum, but is XPS survey spectrum!
  • The authors used TiCl4 for ALD. Is there Cl in the films?

The only thing that can be said from the presented XPS data is that there is titanium on the surface of the sample (we do not know to what extent it is oxidized – TiO2, TiO, Ti, or more probably mixed - TiOx), as well as oxygen and carbon (peak around 285).

Thus, the XPS data and figure 5 do not show absolutely any new and important information!!!

Once again I would like to point out that the authors cannot write that they have got TiO2. They have TiOx and it is impossible to say what is equal to “x” without XRD or correct Ti2p and O1s XPS data analysis!!!

Author Response

Dear

Thanks for the second review, which we read carefully. We performed a re-analysis of XPS on specimens with ALD formed (TiO2/com. Nitinol and TiO2/CVC-NiTi rod). Below we send the answers and the necessary additions/improvements: the text in the main articles is coloured red, and new Figures have been added (5-7). The text has been reviewed again by the native speaker Shelagh Hedges. We think now, hopefully, that the article is now suitable for publication in Metals. On behalf of all the authors, I am sending our best regards.

Rebeka Rudolf

Thanks to the authors for revising the article. But they did not answer all the questions. I still have a lot of questions and comments.

The main problems of the article are related to the low level of characterization of the composition and structure of the obtained coatings!!! By the way, reviewer №2 has essentially the same claims.

The systematic XPS analyses were repeated on a high-quality level XPS spectrometer equipped with an Al-monochromatic source.

The authors state that the TiOx films are too thin for XRD. I have a lot of experience in studying titanium oxide films prepared by ALD. In some cases, it is possible to obtain characteristic XRD patterns even for films 10 nm thick. The authors have grown films with a thickness of 50nm. Therefore, I cannot accept such an answer from the authors. The only real problem with XRD measurements is the roughness of the surface (substrate). Therefore, the surface must be well polished before measurement, or polished witnesses must be used.

But the lack of XRD data is not the main disadvantage of the manuscript. The main problem is XPS !!!

  • Authors stated in the revision version of the manuscript that: « If we take this into account, the indicative ratio of peaks intensities between Ti and O, which is around 1:2, indicates that TiO2 was formed indirectly.»

This is totally wrong! Anyone can notice that this ratio is very far from 2/1, but rather closer to 1/1. But this, in fact, is absolutely irrelevant. The intensities in the XPS do not show anything important. Quantitative analysis and ratios of elements can only be obtained from the ratios of areas under the XPS peaks. But even so, you need to take into account the special coefficients that are different for different elements and instruments!!!

New XPS analyses were performed and quantification of the surface composition was performed from the XPS peak intensities, taking into account the relative sensitivity factors provided by the instrument manufacturer (Reference 32: J. F. Moulder, W. F. Stickle, P. E. Sobol, K. D. Bomben, “Handbook of X-Ray Photoelectron Spectroscopy”, Physical Electronics Inc., Eden Prairie, Minnesota, USA, (1995)).

  • The authors stated in the revision version of the manuscript: «In this spectrum there are two main peaks: Ti2p at a binding energy of 469.2 eV and O1s at a binding energy of 530.7 eV.» This cannot be seen from the presented figure 5. Why 469.2 and not 469.9! Or even 490!!! The scale of the survey spectrum is such that it is absolutely impossible to distinguish it. I have already asked the authors to show individual spectra of Ti2p, O1s, C1s levels, but my request was ignored. Also, what was the measurement interval? 0.1 eV? I believe that the step (interval) was much larger.

New high-energy resolution XPS spectra Ti 2p, O 1s and C 1s were acquired and presented. New XPS spectra were taken with pass energy of 29 eV, energy resolution of 0.6 eV and energy step of 0.1 eV.

  • Also, if you are present the spectrum in a scientific paper, then all peaks must be identified and signed. Unfortunately, the authors only signed Ti2p and O1s. For example, I can clearly see the carbon peak at 285 eV. But the authors do not mention it

The new XPS spectra were corrected for these mistakes.

  • In addition, the spectrum shown in figure 5 is not an XPS spectrum, but is XPS survey spectrum!
  • The authors used TiCl4 for ALD. Is there Cl in the films?

In the new XPS spectra a presence of Cl was checked for carefully, and Cl was not detected.

The only thing that can be said from the presented XPS data is that there is titanium on the surface of the sample (we do not know to what extent it is oxidized – TiO2, TiO, Ti, or more probably mixed - TiOx), as well as oxygen and carbon (peak around 285).

Thus, the XPS data and figure 5 do not show absolutely any new and important information!!!

Once again I would like to point out that the authors cannot write that they have got TiO2. They have TiOx and it is impossible to say what is equal to “x” without XRD or correct Ti2p and O1s XPS data analysis!!!

The new set of XPS spectra show binding energy of the Ti 2p3/2 peak at 458.6 eV, O 1s peak at 530.0 eV and O/Ti ratio 2.5, which altogether confirm the presence of the TiO2 structure. 

Reviewer 2 Report

The authors have made attempts to improve the presentation quality in the paper manuscript. One has to recognize that the quality of the materials analysis in terms of structure and composition remains merely satisfactory, far from very good. The estimation of the film stoichiometry based on XPS peak intensities, somewhat unspecified in this paper, is really a somewhat more complicated procedure. Also it can hardly be understood why the authors say that the TiO2 layers were „extremely thin“. There is nothing extremely thin here in this study, a 50 nm thick TiO2 is actually quite a thick oxide layer whereas XPS analysis depth without additional sputtering extends over some nanometers only.

owveNevertheless, the value of the present study evidently stems from the electrochemical investigations of the anticorrosion properties of the titanium oxide overlayer on nitinol. Therefore the study does add to the existing knowledge and may become published in its current state.

Author Response

Dear

Thanks for the review, which we read carefully. Thanks also for the recommendation and acceptance of our study for publication in Metals. On behalf of all the authors, I am sending best regards.

Rebeka Rudolf

Reviewer 3 Report

It is acceptable now.

Author Response

(The authors gave the same response as above.)

Reviewer 4 Report

I thank the authors that have tried to respond to each of the requirements, but there are still a few issues that need to be addressed before publication.

In figure 7 b the same colors for the substrate had to be kept as in Figure 7a. Tafel diagrams must be represented on a semi-logarithmic scale. But I understand that this cannot be done.

In Figure 8 the diagram for 92 hours -8 days must be added.

The errors obtained when fitting the EIS experimental data must be entered because only in this way we can know if the proposed equivalent circuits are correct or not.

Author Response

Dear!

Thanks for the review, which we read carefully. Below we send the answers and the necessary additions/improvements. The text has been reviewed again by the native speaker Shelagh Hedges. We

think now, hopefully, that the article is now suitable for publication in Metals. On behalf of all the authors, I am sending best regards.

Rebeka Rudolf

I thank the authors that have tried to respond to each of the requirements, but there are still a few issues that need to be addressed before publication.

In figure 7 b the same colors for the substrate had to be kept as in Figure 7a. Improved. Tafel diagrams must be represented on a semi-logarithmic scale. But I understand that this cannot be done.

In Figure 8 the diagram for 192 hours -8 days must be added. Added.

The errors obtained when fitting the EIS experimental data must be entered because only in this way we can know if the proposed equivalent circuits are correct or not.

Added: Line 335: The error obtained when fitting the EIS experimental data was below 0,02 % for all the samples.

Round 3

Reviewer 1 Report

The authors answered all questions and revised the manuscript. Despite the presence of minor flaws, it can be accepted in its present form. 

Please check the manuscript for typos. There are a lot of them.